# Challenges in Maternal and Child Health Services Delivery and Access during Pandemics or Public Health Disasters in Low-and Middle-Income Countries: A Systematic Review

**DOI:** 10.3390/healthcare9070828

**Published:** 2021-06-30

**Authors:** Krushna Chandra Sahoo, Sapna Negi, Kripalini Patel, Bijaya Kumar Mishra, Subrata Kumar Palo, Sanghamitra Pati

**Affiliations:** Regional Medical Research Centre, Indian Council of Medical Research, Bhubaneswar 751023, India; sahookrushna@yahoo.com (K.C.S.); sapnanegi512@gmail.com (S.N.); kripalinie.patel@gmail.com (K.P.); bijaydrster@gmail.com (B.K.M.); drpalsubrat@gmail.com (S.K.P.)

**Keywords:** COVID-19, pandemics, disasters, maternal and child health, reproductive health

## Abstract

Maternal and child health (MCH) has been a global priority for many decades and is an essential public health service. Ensuring seamless delivery is vital for desirable MCH outcomes. This systematic review outlined the challenges in accessing and continuing MCH services during public health emergencies—pandemics and disasters. A comprehensive search approach was built based on keywords and MeSH terms relevant to ‘MCH services’ and ‘pandemics/disasters’. The online repositories Medline, CINAHL, Psyc INFO, and Epistemonikos were searched for studies. We included twenty studies—seven were on the Ebola outbreak, two on the Zika virus, five related to COVID-19, five on disasters, and one related to conflict situations. The findings indicate the potential impact of emergencies on MCH services. Low utilization and access to services have been described as common challenges. The unavailability of personal safety equipment and fear of infection were primary factors that affected service delivery. The available evidence, though limited, indicates the significant effect of disasters and pandemics on MCH. However, more primary in-depth studies are needed to understand better the overall impact of emergencies, especially the COVID-19 pandemic, on MCH. Our synthesis offers valuable insights to policymakers on ensuring the uninterrupted provision of MCH services during an emergency.

## 1. Introduction

Maternal and child health (MCH) is a global priority that has been continually discussed for many decades; it is one of the essential public health services [1,2]. According to a study by the United Nations Interagency Group, 295 thousand maternal deaths per year were estimated in 2017, and there were 18 neonatal deaths per 1000 live births worldwide in 2018 [2]. Sub-Saharan Africa and Southern Asia account for about 86% of the reported maternal deaths worldwide [2,3]. The Sustainable Development Goals (SDGs) provided the target of achieving a global maternal mortality rate of less than 70 per 100,000 live births by 2030, but poor maternal and child health remains a significant challenge in many countries [4].

Although maternal and infant deaths declined substantially from 1990 to 2015, there has been a disproportionate level of health inequity worldwide [2,3]. Children in low-income countries are almost 18 times more likely to die before age five than children in high-income countries; most maternal deaths happen during or immediately after childbirth [4]. Most maternal deaths are due to excess bleeding, high blood pressure, prolonged labor and illegal abortions. The major causes of neonatal death are preterm delivery, extreme infections and birth asphyxia [5]. Women and children survive and thrive in countries that provide everyone with secure, accessible and high-quality health services [6]. Improved health care and a balanced diet are essential contributors to improved MCH services. The lack of essential health condition measures also lead to maternal and infant morbidity and mortality [7]. Until 2019, there was remarkable progress in the MCH areas. However, the rate of development is inadequate to meet the SDGs [4].

Humanitarian emergencies, such as pandemics and disasters, cause an unprecedented disruption in the provision of routine health services. Moreover, the outbreak of COVID-19—declared as a global Public Health Emergency [8,9]—disrupted healthcare services, including that for the most vulnerable communities, such as children and pregnant women, both clinically and socially [10,11,12]. Countries struggling with the pandemic situation redirect both human and material resources to response efforts that lead to inadequate delivery of essential health services; in particular, resource-poor settings are severely affected [7,13]. Therefore, this study aimed to summarize the evidence provided by selected articles through a systematic review of qualitative studies. This study narrated the challenges in obtaining and managing MCH services during pandemics and disasters, which may help to plan a highly resilient health care delivery system to manage MCH services during an emergency in low- and middle-income countries (LMICs).

## 2. Materials and Methods

### 2.1. Narrative Review

This narrative review is registered with the PROSPERO International prospective register of systematic reviews (Registration No: CRD42020184642).

### 2.2. Search Strategy and Selection Criteria

We built a comprehensive search approach based on keywords and MeSH terminology relevant to ‘maternal and child health services’, ‘pandemics’, and ‘disasters’. Three authors independently searched four online repositories—Medline, CINAHL, PsycINFO, and Epistemonikos—to find qualifying studies. We reviewed the retrieved articles in two steps: title and abstract screening, and full-text screening, using inclusion and exclusion criteria. Three authors (S.N., K.P. and K.C.S.) separately reviewed the articles. Following the title and the abstract screening, the potentially relevant studies were evaluated and assessed for eligibility through full-text screening. The reference list of chosen papers was then furthermore searched, and we retrieved related articles. We settled disagreements between authors by discussion and mutual consensus.

We only included studies that identified the problems of maternal and child health services during emergencies. Studies considered eligible for inclusion were limited to the following: published in English, qualitative research articles or qualitative findings of mixed-method research, and studies involving any mass emergencies such as pandemics and disasters. We removed duplicate articles. We excluded articles relating to family planning services and adolescent pregnancy and editorials, review articles, and case reports. We considered articles published until December 2020.

### 2.3. Data Extraction, Synthesis, and Analysis

Two authors (S.N. and K.C.S.) independently extracted the information, and the other authors cross-checked it. We extracted the data for each article into a pre-formed data extraction sheet under the following parameters: study setting, type of emergency (pandemic or disaster) and participants, method of data collection and data analysis, and significant perceived outcomes. The challenges about maternal and child health services, as reported in the studies, were analyzed systematically.

The thematic framework analysis was adopted for data synthesis [14]. We used five phases of framework synthesis approaches for data synthesis. The authors K.C.S. and S.N. were familiar with the review’s objectives and noted recurrent themes throughout the studies. We then identified a thematic framework based on the emerging theme. Three reviewers (S.N., K.P. and K.C.S.) independently read the extracted information to search for themes under a predetermined thematic framework and additional emerging themes. We did the data coding based on the identified themes with MAXQDA Version 18.2.4 (8 April 2020) (VERBI Software, Berlin, Germany). Each primary study was indexed, using framework-related codes. The reviewers sorted the data according to the themes and presented the themes in the analysis table.

### 2.4. Quality Assessment

We used the Consolidated Criteria for the Reporting of Qualitative Research (COREQ) Assessment Tool to assess selected articles’ quality [15]. It is an explicit and comprehensive checklist of 32 items in three fields: research team and reflexivity, study design, and findings analysis.

## 3. Results

### 3.1. Study Selection and Characteristics

We identified a total of 608 articles. Following the title and the abstract screening, we selected 86 potentially relevant articles for a full-text review. After the first round of the full-text review, a total of 46 articles were eligible for inclusion. Of the 46 articles that followed the second round of the full-text review, we excluded 30 articles and included 16 in the study. The cross-reference of the retrieved studies sought four relevant studies. Finally, we selected 20 papers in the narrative review following PRISMA guidelines (Figure 1).

We provided the detailed characteristics of the selected studies in Table 1. Most of the studies were conducted in low- and middle-income countries (LMICs). The primary data collection methods were in-depth interviews (IDIs) and focus group discussions (FGDs). The data were analyzed either by thematic or content analysis approach. Out of 20 studies, seven reported on experience related to the Ebola epidemic, five on COVID-19, two related to the Zika virus, five related to disasters, and one about the conflict situation.

The significant perceived challenges in maternal and child health services during pandemics or disasters are presented in Table 2. The major derived themes comprise (1) maternal health services during an emergency, and (2) child health services during an emergency. The magnitude of the perceived challenges for maternal and child health services during pandemics or disasters are presented in Table 3.

### 3.2. Theme 1: Maternal Health Services during an Emergency

#### 3.2.1. Antenatal Check-Up

All studies indicated the challenges on the use of antenatal services during pandemics and disasters. Poor access to specialists, shortage of medical facilities, out-of-pocket expenditure, and inadequate knowledge about the pandemic were seen as barriers to antenatal care. Many studies suggested the need for satellite clinics to continue antenatal services during disasters. Many studies noticed unbalanced dietary patterns both in pandemics and disasters [16,17,18,19,20,27,35].

Many pregnant women experienced abdominal pain, genital bleeding, urinary tract infections, and stress. However, they did not receive any medical attention [22,32,34]. Several studies documented the lack of transport services or the long way to walk to reach the facilities due to poor road connectivity and the demand for higher transport charges. Another factor identified was the lack of access to specialists and diagnostic facilities in healthcare facilities. The studies reported that during the disaster, healthcare facilities were either damaged or closed [16,19,22,32,34]. Moreover, even traditional birth attendants were scarcely available to provide services [16,34].

“Transportation is a major barrier to maternal care during the flood, as public health facilities or hospitals in rural areas are closed, and health care providers are not available” [16].

Many studies reported affordability as the common factor in both pandemic and disaster situations [16,19,23,35]. Studies indicated that poor financial situation prevented women from using antenatal services. While a few women have received care, they had to pay extra money for specialized advice and diagnostic services. Paying for free check-ups or free medications was also described as the main obstacle [23,30]. Acceptability was another barrier to the use of antenatal care, especially during outbreaks of disease. People thought that they might get an infection when attending health care facilities, traveling by public transport, contacting service providers or an unknown person [17,18,20,21,24,28,33. Few women expressed their fear of transmitting the disease to their fetus and perceived negative impacts, such as consequences related to miscarriage, congenital or genetic abnormalities, and low fetal intelligence [18].

“My biggest worry was that if I get infected, my baby will get it too, it will hurt my baby, and I may have a miscarriage” [18].

Several pregnant women were not willing to receive antenatal care at the hospital. They perceived that the information provided by health officials in the event of a pandemic confused them, as it did not contain much scientific knowledge [23,25]. Gender insensitivity—services provided by male health workers during the post-disaster period—was also a reason for the non-acceptability of services during the disaster [34].

“Health workers were mostly men; we were uncomfortable discussing reproductive health issues with them” [34].

#### 3.2.2. Delivery and Post-Natal Care

Several studies reported an increase in the number of home deliveries by traditional birth attendants (TBAs) or informal health care providers during pandemics and disasters [16,19,27,28,33,35]. The main reason given was that hospital accessibility was difficult due to poor road connectivity and lack of transportation services [16,19,28]. During labor, pregnant women were transported to health facilities by stretchers, baskets, or boats; they spent a large amount of money in this process. Expenditures for transport, food, medicines, and blood transfusion contributed to high, out-of-pocket costs. One study reported that health workers were even asking for bribes in emergencies [29].

“Traditional Birth Assistants were the only option for delivery services during an emergency” [35].

Participants perceived that hospital delivery during the pandemic was risky and expensive; therefore, many women opted for home delivery. Many studies reported that if TBAs or local health care providers were unable to manage obstetric complications, women were referred to public health facilities. Sometimes, ignorance of TBAs or family members caused difficulties, leading to possible maternal deaths [16,20,35]. There has been an increase in non-institutional delivery due to the lack of access to health services in public health facilities. As a result, women were forced to receive benefits from private facilities at higher costs [27].

“During COVID-19, there were no maternity services in public facilities, so they opted to seek the services at the private facilities with higher cost, which was not affordable by many poor women” [27].

“Traditional healthcare providers always convince patients to use their medicines and deliveries at home. They refer in case of complications” [35].

In disaster-affected areas, healthcare providers in nearby health facilities and even TBAs were often unavailable to deliver, resulting in excessive bleeding and maternal death, due to unsafe delivery practices [16,20,35]. Some studies have reported that, although training on infection prevention was provided to health personnel during pandemics, the supply of personal protective equipment (PPE) has been interrupted. In addition, TBAs and nurses were more afraid of treating patients because few had not received training and PPE [20,24,28,33].

Another reason for increased home delivery during disease outbreaks was the risk perception of getting the healthcare staff’s infection [17,24,27,33,35]. Community members were afraid to seek treatment from healthcare facilities. As a result, women avoided hospital delivery. Additionally, accompanying persons were not allowed to stay in the hospital during such time, which created a fear of being left alone [26,33,35].

“Women believed that healthcare workers were injecting patients [with Ebola], so they were scared to come.” [24].

Those who attended health facilities reported negative experiences, such as low awareness, lack of communication, and scarcity of necessary facilities in hospitals. Some studies described that certain drugs that are required during delivery were out of stock. There was also a lack of infrastructure and equipment in hospitals [22,24,28,29]. Healthcare providers perceived that the extreme shortage of healthcare workers in hospitals led to a high workload, resulting in frustration among staff [21,35]. However, few studies reported positive perspectives on health workers’ roles and responsibilities and community health workers. With the support of the government, they acknowledged and carried out their work [22,26]. Many studies documented that the lack of planning and coordination in the health care system was the main reason for the poor condition of health centers [16,34]. One study reported that women were hardly monitored after delivery in disaster-affected regions. During the pandemic, to prevent infection transmission, usually, postnatal services for mothers and neonates are avoided [20,33]. However, a few services are provided virtually, such as breastfeeding and contraceptive-related services [18,33]. One study related to the pandemic identified paying for free services as a significant cause of non-use of postnatal care.

“The planning for post-disaster reproductive health services? There were no such plans” [34].

### 3.3. Theme 2: Child Health Services during an Emergency

#### 3.3.1. Immunization Services for the Child

Only a few studies have documented the challenges of immunization encountered during disasters and pandemics situations. Studies found that Immunization became irregular and declined during disasters and pandemics.

Due to the lack of transport of vaccines, the vaccines being out of stock in stores or depots was a vital issue during disasters [29]. The lack of electricity was a significant problem in the management of the cold chain. Furthermore, challenges were the unavailability of immunization staff and damage to road connectivity between the vaccine storage point and the delivery point. Usually, many children missed routine vaccinations as scheduled, and waited for a month or even until the situation was normal to complete their regular immunization schedule [29,31].

“My child was not vaccinated because of stock-out. We missed a chance to get the vaccination, and they told us to wait another year” [29].

“Medical workers in Immunization clinics are markedly reduced” [31].

At times, during the pandemic, there was sufficient stock of vaccines as per the requirement. Service provided at the point of delivery was as per routine; however, women often did not prefer to visit the immunization site, due to fear or suspicion of infection. Some studies reported perceptions about contracting the infection among children through injections in healthcare facilities [21,24,31]. In some instances, the unavailability of service providers also hindered the provision of vaccination services. Few women also paid for free services. The reduction of outreach services by community health workers also contributed to inadequate coverage of services. However, the vaccination programs did not impair specific settings where they provided immunization services at the doorstep [20,21].

“We avoid immunization for fear of the ‘needles might be injected with Ebola” [21].

#### 3.3.2. Management of Sick Infants

Many studies reported that both disasters and pandemics had an impact on children’s physical and mental health. Several studies reported that diarrhea, common cold, fever, skin disorders, poor appetite, and malnutrition are identified as common disorders among children during disasters [16,22]. Some of the studies also reported physical injuries and psychological trauma to children.

Accessibility to health care is a significant challenge documented in treating sick infants during a disaster, due to severely disrupted health facilities, poor road connectivity during disasters. Delays in care were frequently observed. Several women were forced to seek care either from a local pharmacy shop or from traditional healers, as they were easily accessible. Children were brought to health facilities only when there was an emergency. Several women also reported resorting to home-based treatment, such as boiled water or oral rehydration for children. Providing a healthy diet was another challenge that led to a change in diet patterns during disasters [19,29,32].

“Children were at high risk due to increased susceptibility to cold; there was a risk of exposure to snakebite; there was a lack of transportation and medication” [22].

Moreover, the pandemics’ primary challenge was the low acceptability of health services by beneficiaries from either community health workers (CHWs) or health facilities. There was a significant reduction in the use of child health services—medical consultation and hospital admission—due to fear of transmission of infection [21,28]. Mothers thought that availing treatment from traditional healers instead of CHWs and health facilities would be safer. Many mothers hid children’s illnesses in order to escape referral to higher facilities due to fear of being COVID-19 diagnosed. However, there was an increase in the use of services due to community awareness among the community by religious leaders and health workers [20,21,28]. Another challenge described was the affordability of drugs. Some studies reported that several drugs were out of stock [28].

“I don’t want to take my child to the hospital as I was afraid; they might say that my child has COVID-19” [17].

“The kids were suffering from fever, and their parents hide the information because they thought they would refer to health facilities” [28].

## 4. Discussion

The narrative review reveals that humanitarian emergencies have a potential effect on MCH services. Low utilization and access to maternal health services are described as a common challenge in emergencies. Many women did not receive timely healthcare, resulting in maternal morbidity and mortality [23]. Lack of personal safety equipment in hospitals, fear of infection, and lack of infection prevention training for staff affected the safe delivery of services [32,35].

The gender dimension emerged as a visible barrier to service utilization, especially during disasters [34]. Provisions such as the allocation of labor rooms in flood shelters, the distribution of delivery kits to midwives, and the training of pregnant women in self-care, particularly in disaster-prone areas are essential. Obstetricians and gynecologists should be involved in disaster relief, as they can deal with pregnancy and labor-related complications. A report from Nepal emphasized that female community health volunteers helped alleviate the earthquake’s impact by offering various community services, such as the MCH service [22]. Our findings illustrate the need to improve gender-sensitive policies in providing health services to women in disaster-affected areas.

Acceptability of health care from public health facilities and health staff is a significant problem during pandemics. The Ebola-related literature suggests that, as health facilities became Ebola hubs, mistrust in the general public increased, resulting in a decline in services’ uptake. Previous studies reported a decrease in antenatal services and the institutional delivery rate during the pandemic [36], signifying that communication gaps among various stakeholders influence MCH service access during pandemics [23].

Lack of access to evidence-based information leads to the spread of false news or rumors, resulting in a decline in utilization of services from public health facilities [25]. Pandemic-related studies show that it was not the lack of health service provision during the pandemic. However, the community’s low uptake of health services was reduced to a certain extent during pandemics [36], which indicates that though information access through social sources is imperative, it leads to disseminating false rumors. The engagement of community health workers and community leaders can be crucial in community mobilization during an emergency. This review indicates that there is a need for further study on community participation in emergency service provision.

We found limited studies reporting experience of immunization and pediatric hospitalization during mass emergencies. A study conducted in India showed a probability of a 9–18% rise in children’s acute illness, a 7% increase in malnutrition, and an approximately 18% fall in full immunization of children in disaster-affected areas [37]. Inaccessibility to prescription medicine compels people to take medication from local pharmacies or unauthorized healthcare providers. Missing out on vaccination exposes children to the hazard of vaccine-preventable diseases, which can trigger an inevitable surge of infectious diseases [36]. Therefore, organizing a mass vaccination campaign or catch-up campaign must be a priority in the post-emergency period.

Studies also revealed that women and children are more vulnerable to psychiatric illnesses due to emergencies [38,39]. A systematic review showed that women’s mental health affects child development more after the disaster than during the disaster itself [40]. Hence, psychological services must be made available to prevent psychological illness among women and children.

We present the quality assessment of the studies, using the COREQ checklist in Table 4. Many studies presented interviewer credentials and interviewer relationship with the participants. Most of the articles provided detailed theoretical frameworks and participants’ recruitment strategies. All the studies followed the standard analysis and reporting guidelines. None of the studies conducted repeat interviews, and few studies documented non-participation of the participants and member-check approaches. Limited numbers of studies mentioned the duration of interviews and data saturation during the interview. Six out of 20 studies took field notes while conducting IDIs or FGDs. Half of the studies mentioned the number of data coders. Out of 20, only eight studies used software for the data coding and compiling. However, only seven studies described and presented the detail-coding tree.

### 4.1. Implication for Practice and Policy

In health emergencies, the implementation of global and national child health interventions, such as Integrated Management of Childhood Illness (IMCI), is often a concern due to the lack of an emergency preparedness plan. Emergency service delivery techniques, such as modules, guidelines, and capacity building, can be included. Services such as health-related awareness and community engagement can be helpful in emergencies. Furthermore, there should be compensation or reprioritization of community health workers’ duties to fulfill the immunization, maternal, post-natal, nutritional needs, and psychological needs of women and children in emergencies.

### 4.2. Methodological Considerations

Although we systematically reviewed studies relevant to MCH services, we omitted the studies dealing with family planning services. We tried to use all possible terms for the creation of a search strategy. However, due to language restrictions, the search strategy was limited to the articles published in English. We skipped the related articles published in other languages. This study’s authors have varied backgrounds, including clinical nursing, medical sociology, community medicine, and obstetrics and gynecology—all with a public health perspective.

## 5. Conclusions

The evidence suggests that the severity of the impact of disasters and pandemics on MCH has been significant; however, this review indicates the need for more primary qualitative research to understand better the overall effect of emergencies on mother and child health and wellbeing. Our study provides the first ever indicative evidence for policymakers to establish priority interventions and ensure the continuous provision of MCH services, such as prenatal care, safe birth, post-natal care, and safe childhood, in emergencies.

## Figures and Tables

**Figure 1 healthcare-09-00828-f001:**
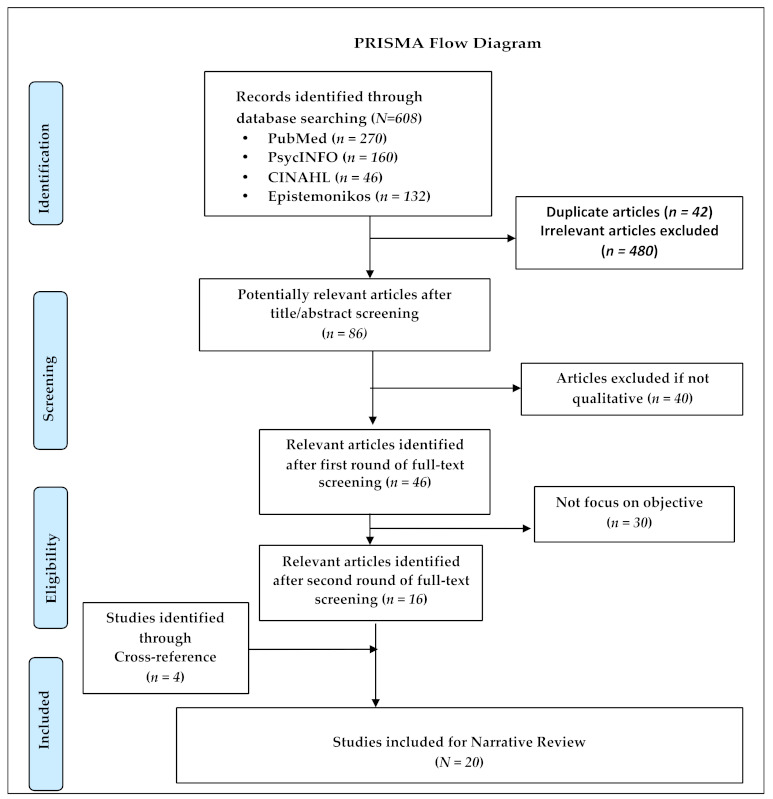
PRISMA flow diagram.

**Table 1 healthcare-09-00828-t001:** Characteristics of the selected studies.

Author	Setting	Pandemic/Disaster	Study Participants	Data Collection Method	Analysis Method	Major Topic Discussed
Abdullah et al., 2019 [16]	Bangladesh	Flood	Healthcare Providers, Pregnant and postnatal women	Focus group discussions (*n* = 3), In-depth interviews (*n* = 8)	Content analysis	Maternal care
Aridi et al., 2020 [17]	Kenya	COVID-19	Postnatal women	Telephonic interviews (*n* = 71)	Thematic analysis	Access to MCH services
Bakouei et al., 2020 [18]	Iran	COVID-19	Pregnant women	Telephonic interviews (*n* = 12)	Content analysis	Pregnancy experience
Brunson, 2017 [19]	Nepal	Earthquake	Women	In-depth interviews (*n* = 14)	Thematic analysis	Maternal and Child Health
Dynes et al., 2015 [20]	Sierra Leone	Ebola	Healthcare workers, pregnant and lactating women	Focus group discussions (*n* = 9)	Content analysis	Antenatal care, Postnatal care and Immunization
Elston et al., 2015 [21]	Sierra Leone	Ebola outbreak	Local stakeholders, Community Health Workers, and Social mobilizers	Focus group discussions (*n* = 7), In-depth interviews (*n* = 60)	Thematic analysis	Maternal and Child Health
Fredricks et al., 2017 [22]	Nepal	Earthquake	Key Informants, Community Health Workers	Focus group discussions (*n* = 2), In-depth interviews (*n* = 17)	Thematic analysis	Maternal and Child Health
Gomez et al., 2020 [23]	Colombia	Zika virus	Women	In-depth interviews (*n* = 6)	Thematic analysis	Prenatal services
Jones et al., 2017 [24]	Sierra Leone	Ebola	Midwives, Medical staffs and Program Managers	In-depth interviews (*n* = 66)	Framework analysis	Maternal and Child Health
Linde-Arias et al., 2020 [25]	Brazil, Puerto Rico	Zika virus	Women	In-depth interviews (*n* = 24)	Thematic analysis	Social effects of pandemic on Maternal health
Lori et al., 2017 [26]	Liberia	Ebola	Traditional Birth Attendants, Certified midwives, Women	In-depth interviews (*n* = 21)	Content analysis	Maternal and Child Health
Lusambili et al., 2020 [27]	Kenya	COVID-19	Healthcare staff, antenatal and postnatal refugees women	In-depth interviews (*n* = 25)	Thematic analysis	Barriers of MCH services utilization
Miller et al., 2018 [28]	Liberia	Ebola outbreak	Stakeholders	Focus group discussions (*n* = 16), in-depth interviews (*n* = 44)	Thematic analysis	Maternal and Child Health
Nidzvetska et al., 2017 [29]	Ukraine	Conflict	Mothers	In-depth interviews (*n* = 9)	Thematic analysis	Maternal and Child Health
Pieterse and Lodge, 2018 [30]	Sierra Leone	Ebola	Healthcare Providers	Focus group discussions (*n* = 3), In-depth interviews (*n* = 25)	Thematic analysis	Maternal and Child Health
Saso et al., 2020 [31]	Multi-countries	COVID-19	Members of IMPRINT	Online survey (*n* = 48)	Thematic analysis	Immunization
Sato et al., 2016 [32]	Yolanda	Typhoon	Women	Focus group discussions (*n* = 4)	Content analysis	Maternal and Child Health
Semaaan et al., 2020 [33]	Global	COVID-19	Healthcare professionals	Online survey (*n* = 714)	Thematic analysis	Maternity care
Sohrabizadeh et al., 2018 [34]	Iran	Disasters	Health workers and Experts	In-depth interviews (*n* = 22)	Content analysis	Maternal and Child Health
Theuring et al., 2018 [35]	Sierra Leone	Ebola	Providers, pregnant and postnatal women	Focus group discussions (*n* = 6)	Content analysis	Maternal and Child Health

**Table 2 healthcare-09-00828-t002:** Major challenges in maternal and child health services during pandemic or disaster.

Maternal Health Services	Emergency Situation	
	Pandemic	Disaster
Antenatal check-up	Poor access to specialists Unavailability of diagnostic services Out of pocket payment to healthcare providers Inadequate scientific information Hastened health services Virtual care Hesitant to visit Long waiting time	Poor transportation services Unavailability of specialists No satellite clinics Closed health facilities Unbalanced nutrition practices Post-disaster services by male health workers
Delivery and Post Natal Care	Unavailability of personal protective equipment No training of staff on infection prevention Rumors—staff injecting infection Unfavorable working attitudes of staffs Lack of basic facilities at hospital Shortages of drugs, instruments or other supplies Understaffed facilities Ban on support companion Increased home deliveries Inclination to private clinics Reduced follow-ups Payment for free care	Traditional birth attendant only accessible option No ambulance services Indirect expenses in hospital No place for delivery Unavailability of specialists Unsafe delivery practices No planning for post disaster services Difficulty obtaining baby formula Lack of follow-ups
**Child Health Services**		
Immunization services for child	Reduction in immunization services Fear of needles injecting disease Reduction in the outreach services Paying for vaccination records Anti-vaccine sentiment	No vaccines in stock Delayed arrival of vaccines stock Payment for free services
Management of sick infants	Reduced consultation Reduction in pediatrics admission Paid for free medications Ambiguity of referral criteria Hiding illness of children Rumors regarding infection transmission Fear of specialized treatment units Increased use of telemedicine	Diarrheal deaths and other common illness Psychological trauma Poor accessibility to healthcare facilities Treatment from small pharmacies Inadequate mental health support Shortage of drugs Poor nutrition

**Table 3 healthcare-09-00828-t003:** Magnitude of the perceived challenges for maternal and child health services during pandemic or disasters.

MCH Services	Pandemic (*n* = 14)				Disaster (*n* = 6)			
	Accessibility	Availability	Affordability	Acceptability	Accessibility	Availability	Affordability	Acceptability
Maternal Health services								
Diagnostic services	SC	SC	SC	VC	VC	SC	NR	NR
Doctor consultation	SC	SC	SC	VC	VC	SC	VC	NC
Transportation	SC	SC	SC	VC	VC	VC	SC	NR
Drugs and consumables	SC	SC	SC	VC	SC	SC	SC	NC
Labor room/intra-natal	NR	NC	SC	VC	SC	SC	NR	NR
Hospital stay	NR	NC	SC	VC	SC	SC	SC	NR
Child health services								
Immunization	NR	SC	SC	VC	SC	NR	NR	NR
Doctor consultation	NR	NR	SC	VC	SC	VC	NR	NR
Transportation	NR	NR	NR	SC	SC	VC	NR	NR
Drugs and consumables	NR	SC	SC	SC	SC	VC	NR	NC
Diagnostic services	NR	NR	NR	SC	SC	SC	SC	NR
Hospital stay	NR	NR	NR	VC	SC	SC	NR	NR

Not Reported (NR), No Challenge (NC), Somewhat Challenge (SC), Very much Challenges (VC).

**Table 4 healthcare-09-00828-t004:** Quality assessment of the studies, using consolidated criteria for reporting qualitative research (COREQ) assessment tool.

Domains	Abdullah et al., 2019 [16]	Aridi et al., 2020 [17]	Bakouei et al., 2020 [18]	Brunson, 2017 [19]	Dynes et al., 2015 [20]	Elston et al., 2015 [21]	Fredricks et al., 2017 [22]	Gomez et al., 2020 [23]	Jones et al., 2017 [24]	Linde-Arias et al., 2020 [25]	Lori et al., 2017 [26]	Lusambili et al., 2020 [27]	Miller et al., 2018 [28]	Nidzvetska et al., 2017 [29]	Pieterse and Lodge, 2018 [30]	Saso et al., 2020 [31]	Sato et al., 2016 [32]	Semaan et al., 2020 [33]	Sohrabizadeh et al., 2018 [34]	Theuring et al., 2018 [35]
Research team and reflexivity																				
*Personal characteristics*																				
Interviewer	●	●	●	●	●	●	●	●	●	●	●	●	●	●	●	●	●	●	●	●
Credentials	●	●	●	●	●	●	●	●	●	×	●	●	●	●	●	●	●	●	●	●
Occupation	×	●	●	×	×	●	×	×	●	×	●	●	×	×	×	●	●	●	×	×
Gender	×	×	×	×	×	×	×	×	×	×	×	×	×	×	×	×	●	×	×	●
Experience and training	×	●	●	×	×	×	×	×	×	×	×	●	●	×	×	●	●	×	×	●
*Relationship with participants*																				
Relationship established	●	●	●	●	●	●	●	●	●	●	●	●	●	●	●	●	●	●	●	●
Participant knowledge of the interviewer	●	●	●	●	●	●	●	●	●	●	●	●	●	●	●	●	●	●	●	●
Interviewer characteristics	●	●	●	●	●	●	●	●	×	●	●	●	●	●	●	×	●	●	●	●
Study design																				
*Theoretical Framework*																				
Methodological orientation	●	●	●	●	●	●	●	●	●	●	●	●	●	●	●	●	●	●	●	●
*Participant selection*																				
Sampling	●	●	●	●	●	●	●	●	●	●	●	●	●	●	●	●	●	●	●	●
Method of approach	●	●	●	●	●	●	●	●	●	●	●	●	●	●	●	●	●	●	●	●
Sample size	●	●	●	●	●	●	●	●	●	●	●	●	●	●	●	●	●	●	●	●
Non-participation	×	●	×	×	×	×	×	●	×	×	×	●	×	●	×	●	×	●	×	×
*Setting*																				
Setting of data collection	●	●	●	●	●	×	●	●	●	●	●	●	●	●	●	●	●	●	●	●
Presence of non-participants	×	×	×	×	×	×	×	●	×	×	×	●	×	×	×	×	×	×	×	×
Description of sample	●	●	●	×	●	×	●	●	●	●	●	●	●	●	●	●	●	●	●	●
*Data collection*																				
Interview guide	●	●	●	●	●	×	●	●	●	●	●	●	●	●	●	●	●	●	●	●
Repeat interviews	×	×	×	×	×	×	×	×	×	×	×	×	×	×	×	×	×	×	×	×
Audio/visual recording	●	●	●	●	×	×	●	●	●	●	●	●	●	●	×	×	●	×	●	●
Field notes	●	×	×	×	●	×	×	●	×	×	×	●	●	×	×	×	×	×	×	●
Duration	●	×	●	●	×	×	×	●	●	●	×	×	×	●	×	●	●	×	●	●
Data saturation	×	●	●	×	×	×	×	×	×	×	●	●	×	●	×	×	●	×	●	●
Transcripts returned	×	●	●	×	×	×	×	×	×	×	×	×	×	×	×	●	●	×	●	×
Analysis and findings																				
Number of data coders	●	●	×	×	×	×	●	×	●	●	●	●	●	×	×	●	×	●	×	●
Descriptions of the coding	×	●	●	×	×	×	×	●	×	●	×	●	×	×	×	●	×	●	×	×
Derivation of themes	●	●	●	×	●	×	●	●	●	●	●	●	●	●	●	●	●	●	●	●
Software	×	●	×	●	×	×	●	●	●	●	×	●	×	×	●	×	×	×	×	×
Participant checking	×	●	●	×	×	×	×	×	×	×	×	×	×	×	×	×	×	×	●	×
*Reporting*																				
Quotations presented	●	●	●	×	×	●	●	●	●	●	●	●	●	●	●	●	●	●	●	●
Data and findings consistent	●	●	●	●	●	●	●	●	●	●	●	●	●	●	●	●	●	●	●	●
Clarity of major themes	●	●	●	●	●	●	●	●	●	●	●	●	●	●	●	●	●	●	●	●
Clarity of minor themes	●	●	●	●	●	●	●	●	●	●	●	●	●	●	●	●	●	●	●	●

● Represents addressed the point, and **×** represents not addressed the points.

## Data Availability

The data presented in this study are available on request from the corresponding author.

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
