# Peer review of "Challenges in Maternal and Child Health Services Delivery and Access during Pandemics or Public Health Disasters in Low-and Middle-Income Countries: A Systematic Review"

_healthcare, 2021, doi:10.3390/healthcare9070828_

Round 1

Reviewer 1 Report

The central topic of the paper is to evaluate through a narrative review the scientific evidence regarding access to primary care during pandemics and public health disasters.

From a methodological point of view, the work is conducted in a rigorous manner by applying the PRISMA guidelines required for systematic reviews.

The results are reported in a clear and readable way.

The discussion also correctly highlights public health implications and methodological considerations.  

Author Response

RC: The central topic of the paper is to evaluate through a narrative review the scientific evidence regarding access to primary care during pandemics and public health disasters. From a methodological point of view, the work is conducted in a rigorous manner by applying the PRISMA guidelines required for systematic reviews.

AR: Thanks a lot for your comments and suggestions.

RC: The results are reported in a clear and readable way. The discussion also correctly highlights public health implications and methodological considerations.

AR: Thank you.  

Reviewer 2 Report

This study aims to summarize challenges in maternal and child health service delivery and access during pandemics or public health disasters. Summarize my thoughts on how this manuscript could be improved:

-The word "narrative" should be removed from the title and just say that it is "a systematic review".

-The title should be more specific and indicate that it is about countries with low development.

- The methodology is correct and the discussion is clear and comprehensive.

Author Response

RC: This study aims to summarize challenges in maternal and child health service delivery and access during pandemics or public health disasters. Summarize my thoughts on how this manuscript could be improved.

AR: Thanks a lot for your comments and suggestions. We revised the manuscript as suggested. The modification/changes in the revised manuscript were highlighted in yellow colour.

RC: The word "narrative" should be removed from the title and just say that it is "a systematic review".

AR: As suggested, we removed the word "narrative" from the title.

RC: The title should be more specific and indicate that it is about countries with low development.

AR: Thank you. We have revised the title accordingly.

RC: The methodology is correct and the discussion is clear and comprehensive.

AR: Thank you. 

Reviewer 3 Report

Thank you for the opportunity to review this work on Maternal and Child Health Services during health disasters and pandemics.
It is a well-structured narrative systematic review.
Just point out a couple of formal aspects: figure 1 must be improved in quality. Include references in the results section when phrases such as "many studies indicate that ...", "studies found ..." are commented. There are only references in verbatim sentences of the studies, but not when their results are named.
This is a very illustrative review of the lack of resources and policies directed at women and children.

Author Response

RC: Thank you for the opportunity to review this work on Maternal and Child Health Services during health disasters and pandemics. It is a well-structured narrative systematic review.

AR: Thanks a lot for your comments and suggestions. We revised the manuscript accordingly. The modification/changes in the revised manuscript were highlighted in yellow colour.

RC: Just point out a couple of formal aspects: figure 1 must be improved in quality.

AR: We have provided better quality figure.

RC: Include references in the results section when phrases such as "many studies indicate that ...", "studies found ..." are commented. There are only references in verbatim sentences of the studies, but not when their results are named.

AR: We have added references in the results section.

RC: This is a very illustrative review of the lack of resources and policies directed at women and children.

AR: Thank you.